

# An adaptive data-driven architecture for mental health care applications

Aishwarya Sundaram[1], Hema Subramaniam[2], Siti Hafizah Ab Hamid[2] and Azmawaty Mohamad Nor[3]

[1] Institute for Advanced Studies, Universiti Malaya, Kuala Lumpur, Malaysia
[2] Department of Software Engineering, Faculty of Computer Science and Information Technology, Universiti Malaya, Kuala Lumpur, Malaysia
[3] Department of Educational Psychology and Counselling, Faculty of Education, Universiti Malaya, Kuala Lumpur, Malaysia

Corresponding authors
Aishwarya Sundaram,
s2029083@siswa.um.edu.my
Hema Subramaniam,
hema@um.edu.my

## ABSTRACT

**Background:** In the current era of rapid technological innovation, our lives are becoming more closely intertwined with digital systems. Consequently, every human action generates a valuable repository of digital data. In this context, data-driven architectures are pivotal for organizing, manipulating, and presenting data to facilitate positive computing through ensemble machine learning models. Moreover, the COVID-19 pandemic underscored a substantial need for a flexible mental health care architecture. This architecture, inclusive of machine learning predictive models, has the potential to benefit a larger population by identifying individuals at a heightened risk of developing various mental disorders.

**Objective:** Therefore, this research aims to create a flexible mental health care architecture that leverages data-driven methodologies and ensemble machine learning models. The objective is to proficiently structure, process, and present data for positive computing. The adaptive data-driven architecture facilitates customized interventions for diverse mental disorders, fostering positive computing. Consequently, improved mental health care outcomes and enhanced accessibility for individuals with varied mental health conditions are anticipated.

**Method:** Following the Preferred Reporting Items for Systematic Reviews and Meta-Analyses guidelines, the researchers conducted a systematic literature review in databases indexed in Web of Science to identify the existing strengths and limitations of software architecture relevant to our adaptive design. The systematic review was registered in PROSPERO (CRD42023444661). Additionally, a mapping process was employed to derive essential paradigms serving as the foundation for the research architectural design. To validate the architecture based on its features, professional experts utilized a Likert scale.

**Results:** Through the review, the authors identified six fundamental paradigms crucial for designing architecture. Leveraging these paradigms, the authors crafted an adaptive data-driven architecture, subsequently validated by professional experts. The validation resulted in a mean score exceeding four for each evaluated feature, confirming the architecture's effectiveness. To further assess the architecture's practical application, a prototype architecture for predicting pandemic anxiety was developed.

# INTRODUCTION

In recent years, there has been an increasing awareness of the significance of mental health and well-being. As mental disorders continue to rise globally (*Cénat et al., 2021*), policymakers, scholars, and health care professionals are actively exploring innovative approaches to provide effective care and support to those in need. An emerging and evolving strategy involves the adoption of data-driven mental health care systems utilizing machine learning algorithms (*Vaishali Narayanrao & Lalitha Surya Kumari, 2020*). These systems aim to leverage extensive information gathered from diverse sources, such as electronic health records, wearable devices, and patient-reported outcomes (*Aggarwal & Girdhar, 2022*; *An et al., 2022*).

Within the realm of data-driven mental health care systems, machine learning models play a pivotal role in uncovering patterns (*Nayan et al., 2022*) that health care practitioners might potentially overlook. A spectrum of machine learning models, including supervised learning, unsupervised learning, and deep learning, has been utilized to scrutinize and comprehend mental health data. For instance, supervised learning models (*Saha et al., 2023*) can undergo training on labeled datasets, enabling predictions of specific outcomes such as the probability of a relapse or the effectiveness of a particular intervention. In contrast, unsupervised learning techniques (*Krishnan, Purohit & Rangwala, 2020*) discern inherent patterns within data without predetermined labels, thereby aiding in the exploration and revelation of novel insights.

In the field of mental health, various data sources come into play, ranging from electronic health records (*Taquet et al., 2021*) and genetic information to brain imaging data and social media activity. Machine learning models adeptly handle and integrate this diverse data, providing a holistic view of an individual's mental health profile. Extracted features from these datasets may encompass socio-demographic information, medical history, lifestyle factors, and behavioral patterns, contributing to a comprehensive understanding of mental health nuances (*Liu et al., 2022*).

A notable advantage of data-driven mental health care systems lies in their ability to provide tailored and precise interventions, a quality accentuated especially during pandemics like COVID-19 (*Arivoli, Golwala & Reddy, 2022*). The remarkable capability of machine learning in mental health is its aptitude to customize interventions according to individual needs. Through the analysis of extensive and intricate datasets, machine learning models can discern personalized treatment plans based on an individual's unique characteristics, thereby augmenting the precision and efficacy of mental health interventions and deviating from conventional one-size-fits-all approaches (*Mukhiya, Lamo & Rabbi, 2022*).

Within the framework of data-driven systems, machine learning algorithms (*Gandhi & Mishra, 2022*) undergo a series of events, commencing with data collection and dataset extraction. Following this, these algorithms oversee diverse tasks, encompassing data
preprocessing, which involves data filtration, handling missing values, noise reduction, tokenization, vectorization, manipulation, data presentation, and final report visualization. This integrated approach underscores the potential of combining machine learning algorithms with data-driven systems to significantly impact mental health care, offering a nuanced and personalized perspective aligned with the evolving landscape of mental health applications (*Alrizq et al., 2022*).

Thus, data-driven systems and machine learning prediction are inherently interconnected (*Thieme, Belgrave & Doherty, 2020*), forming the cornerstone of any decision-support system reliant on valuable data (*Khumprom & Yodo, 2019*). The data within these systems is typically categorized into structured, unstructured, and semi-structured formats (*Siriyasatien et al., 2018*), which are subsequently processed and analyzed. In addition to acquiring data from traditional sources like social networking platforms, *e.g.*, Facebook and Twitter (*Alharbi & Fkih, 2022*), datasets obtained from surveys, cohort studies, and direct face-to-face interviews, among other modalities, crucially depend on passive sensing. This involves collecting data not only from social media but also from ubiquitous mobile devices during routine activities. This multifaceted dataset, including spatial data, sensor data, graphs, and real-time transactions (*Burkom et al., 2021*), plays a pivotal role in training machine learning algorithms. Recognizing the importance of passive sensing within the data collection framework is imperative, given its ability to enhance the understanding of diverse origins contributing to the intricate and dynamic datasets instrumental in decision-making processes within these mental health systems.

Data-driven systems provide user-friendly access, facilitating the integration of diverse data sources, intuitive data manipulation, and varied conceptual reporting of interpreted results. The importance of data-driven systems lies in the reliability of data-driven decision-making. The data-driven decision-making framework encompasses multiple stages, including data collection, organization, analysis, summarization, synthesis, and prioritization (*Yu et al., 2021*). Consequently, the health care domain will persist in depending on data and machine learning models to guide the transformation process, necessitating a dependable and adaptive machine learning data-driven architectural framework (*Aldabbas et al., 2022*). Given that data is collected from various sources through multiple channels and in diverse formats, it becomes crucial to design an interoperable framework that ensures a smooth and secure process flow before the actual manipulation by ensemble models. While accentuating the significance of reusability, automated machine learning prediction, and human decision support (*Alreshidi & Ahmad, 2019*), it is imperative to take into account the explainability feature of these decision support systems, which empower human experts to navigate and interpret the decision-making process (*Terhorst, Knauer & Baumeister, 2023*).

Building upon the necessity for an adaptive machine learning data-driven framework, the systematic literature review (SLR) unfolds across the following chapters. The related work section delves into pertinent studies to identify existing gaps, laying the groundwork for subsequent sections. Following this, the research problem and associated research questions are detailed, elucidating the focal point of the study. The methodology section

outlines the chosen research approach and techniques employed, ensuring transparency in the research process. Subsequently, the results and discussion section presents the findings and conducts a comprehensive analysis to foster a deeper understanding. The case study section provides a specific application of the research, further illustrating its practical implications. Finally, the conclusion synthesizes key insights, reaffirming the study's significance and suggesting potential avenues for future research.

## RELATED WORK

In this section, the authors explore recent studies about the architectural design of health care systems driven by data analytics. The study by *Kaur, Sharma & Mittal (2018)* delved into big data analytics in the health care domain, leading to the development of four crucial pillars: patient-centric care, incorporating health records, drug history, patient behavior, and preferences; real-time patient monitoring through wearable sensors; predictive analysis of diseases; and enhancement of treatment methods. The health care architecture established in this study comprises four layers: data source, storage, security, and a machine learning-based application layer, with a primary focus on security and privacy.

Similarly, *Patel & Gandhi (2018)* concentrated on big data analytics in health care, employing ensemble models. Despite the evident advantages of deploying big data analytics with machine learning, the review underscores key challenges associated with managing diverse data structures, data storage, management, integration, and processing.

The SLR by *Plaza, Diaz & Perez (2018)* focused on cyber-physical systems, integrating sensing, computing, and communication to monitor, control, and interact with physical processes. The primary objective of this systematic review was to identify successful solutions that could serve as valuable guidance for architects and practitioners in their health care projects. The synthesis of the search results generated a knowledge base of software architectures for health care cyber-physical systems, encompassing stakeholders' interests, functional and non-functional requirements, quality aspects, architectural views and styles, and the components of architectural designs.

On the other hand, *Avci, Tekinerdogan & Athanasiadis (2020)* focused on software architectures for extensive data systems, carefully considering various elements such as application domain, architectural viewpoints, patterns, concerns, quality attributes, design methods, technologies, and stakeholders. This SLR thoroughly examined big data software designs, including a survey and comparison of architectures across multiple domains.

A survey and comparison of big data architectures were carried out, covering multiple application domains (*Macak, Ge & Buhnova, 2020*). This study involved selecting representative architectures from each domain to guide researchers and practitioners in their respective fields. Furthermore, a cross-domain comparison was conducted to identify similarities and differences among the domain-specific architectures. The study concluded by presenting practical guidelines to aid big data researchers and practitioners in constructing and improving their architectures, leveraging insights gathered from this research.

The study by *Schymanietz, Jonas & Möslein (2022)* focused on data-driven service innovation, which involves integrating data and analytics into the domain as an analytical
unit. The research involved systematic and expert reviews, with data as the primary source, exploring and synthesizing various attributes and terms related to data science to enhance organizational capabilities.

In another study by *Mukhiya, Lamo & Rabbi (2022)*, a user profiling model for a reference architecture was introduced to adapt and personalize interventions based on individual user needs. Using this proposed reference architecture, an open-source framework for an adaptive intervention design and planning tool was developed. The framework underwent evaluation through a combination of a case study, expert evaluation, and software quality metrics. Factors such as adaptability, scalability, reusability, security, interoperability, and modifiability were assessed. However, it is important to note that the evaluation did not cover other critical metrics, including reliability, data quality, and performance.

In a study by *Khan, Khan & Nazir (2022)* comprehensive and systematic research spanning articles from 2011 to 2021 focused on analyzing the health care domain in disease diagnosis using data analytics. The findings suggested that integrating advanced hybrid machine learning-based models and cloud computing applications could bring several benefits to the health care sector. These advantages include cost reduction in treatments, decreased simulation time, and improved quality of care. Policymakers can promote the adoption of these technologies to encourage researchers and practitioners to develop more sophisticated disease-diagnosing models, ultimately enhancing the overall quality of patient treatment. The study also emphasized that architectures for cognitive computing with hybrid machine learning are essential tools for the data-driven analysis of health care big data, offering promising avenues for the future. Based on the background study, several key findings emerge.

## RESEARCH PROBLEM

The studies in the related work indicate a lack of comprehensive research in investigating data-driven architecture through the lens of machine learning, a vital aspect of mental health care. Furthermore, the rising prevalence of mental disorders highlights the need for an adaptive architecture capable of addressing various mental health conditions to achieve cost and time efficiencies (*Swain & Patra, 2022*; *World Health Organization, 2022*; *Zhou et al., 2021*). The identified research gaps form the basis for the SLR. To bridge these gaps, research questions were formulated using the PICOC model (*Mengist, Soromessa & Legese, 2020*), as outlined in Table 1. The background study and the PICOC model collectively informed the framing of the following research questions:

RQ 1: What are the strengths and limitations of current studies on data-driven architecture?

RQ 2: What are the key essential paradigms (KEP) for designing an adaptive data-driven architecture?

Building upon the insights gained from addressing RQ1, the knowledge synthesized becomes the foundation for tackling RQ2. Understanding the strengths and limitations of existing architecture studies is crucial for formulating the KEPs required for designing

**Table 1  PICOC model.**

| Components | Description |
|---|---|
| Population | Architectural patterns. |
| Intervention | Data-driven health care systems. |
| Comparison | NA. |
| Outcomes | Identifying the strengths and limitations of data-driven systems architecture. |
| Contexts | Review of the strengths and limitations of various architectural patterns employed in data-driven systems. |

adaptive data-driven architectures. This process involves mapping strengths and limitations through four essential steps:

1) Identify strengths and limitations: Review each study individually and identify the strengths and limitations of the data-driven architectural patterns discussed in the study.

2) Thematic categorization: Group together the identified strengths and limitations that have similar implications or characteristics. For example, if multiple studies mention "interoperability" and "exchangeability" as a strength or limitation, categorize them under the theme of "Data service".

3) Frame KEPs: Based on the categorized strengths and limitations, frame KEPs that are critical for a data-driven architectural pattern. These paradigms represent the necessary aspects that should be present in any effective data-driven architecture.

4) Validate and refine: Ensure that the identified KEPs foster the design of data-driven architectural patterns. Validate the mapping with relevant experts to refine and improve the clarity and comprehensiveness of the KEPs.

## METHODOLOGY

The review methodology comprises several deliberate steps. Initially, the authors identified the research problem and questions, drawing insights from existing related work. Subsequently, the authors undertook an extensive literature search and screening process, utilizing the Preferred Reporting Items for Systematic Reviews and Meta-Analyses (PRISMA) framework (*Moher et al., 2009*) to meticulously select studies for inclusion in the SLR (*Newaz, Giggins & Ranasinghe, 2022*). The PRISMA flowchart, illustrated in Fig. 1, depicts the systematic process for selecting articles from the initial stages to the ultimate selection.

The development of the literature search and screening strategy (*Klerings et al., 2023*), along with data extraction, was initially designed by the first two authors. A review meeting involving all four authors was conducted to assess the research design strategy. During this meeting, the remaining two authors provided valuable suggestions and recommendations to address any discrepancies. The third and fourth authors were responsible for data

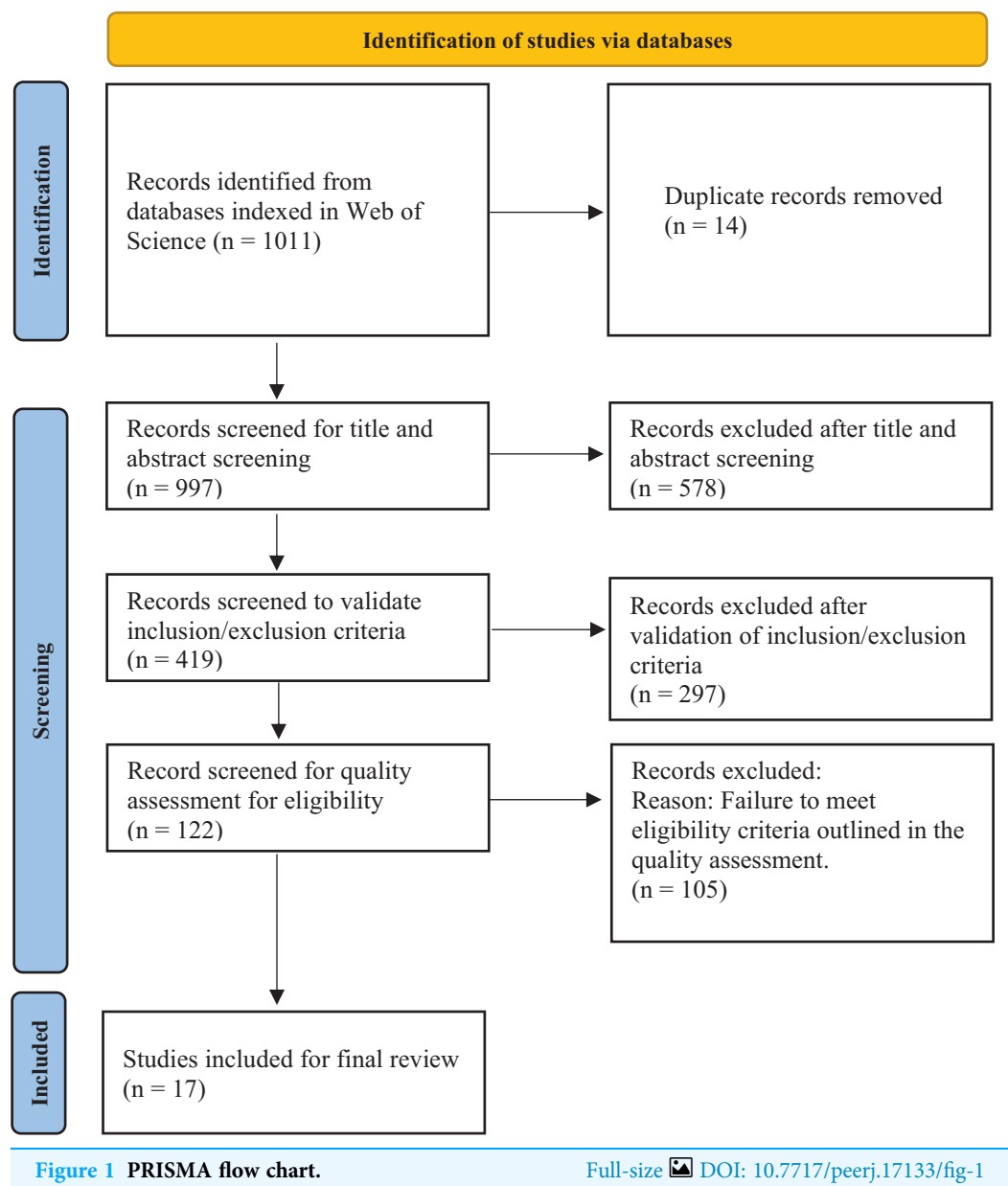

**Figure 1 PRISMA flow chart.**

preparation, analysis, and interpretation. All collected data were meticulously recorded and stored in an Excel spreadsheet to ensure accuracy and transparency.

To minimize bias, the first and second authors independently validated the extracted data. In a subsequent review meeting, all final studies were compared, and each author expressed their agreement or disagreement with the identified results. The authors thoroughly discussed these findings. In cases where differences in opinions arose, the variations were carefully examined and deliberated upon until a consensus was reached among the team. Following this, the results and discussions regarding the selected studies were drafted.

Then, the methodology extends to the design of an adaptive data-driven architecture based on the identified KEPs. This step is crucial because it synthesizes the knowledge gained from the SLR into a tangible and applicable framework, providing a practical solution. The architecture is further validated by subject matter experts from the software development field, ensuring that it aligns with standards and best practices for real-time deployment. The experts demonstrated an average professional experience spanning 10 to 15 years in the software domain, specializing in the architecture and development of diverse projects. Initially, the KEPs design process was explained, with relevant details provided. Subsequently, the adaptive data-driven architecture, derived from the identified KEPs, was deliberated with the experts, offering clear explanations of the components and intended applications. Any queries raised by the experts were comprehensively addressed.

Following this, the Likert scale (*Perez-Benito et al., 2020*) rating methodology was explained, and the experts proceeded with the rating process, including the collection of additional inputs, if any. As a valuable suggestion from the experts to validate the architecture with real-time implementation, a conceptual prototype was implemented from the designed architecture as a first step in a case study tailored for managing mental disorders. This conceptual design serves as a practical testing ground for the proposed architecture, to evaluate its effectiveness and feasibility in real-time deployment for future endeavors. Figure 2 presents a visual representation of the overview of the method employed to conduct our systematic literature review.

## LITERATURE SEARCH AND SCREENING

The process of conducting literature search and screening involved various steps such as selecting appropriate data sources, formulating search strings, setting up inclusion and exclusion criteria, and creating a checklist for quality assessment to determine the final studies for review. The objective is to include a diverse range of databases, spanning studies from mental health in psychology to software architecture design in software engineering, to assemble pertinent articles for the study. Hence, the authors have chosen Web of Science indexing, which includes a wide range of multiple databases covering both mental health and software architecture. Its reliability is supported by comprehensive coverage of databases, strict publishing standards, meticulous quality control procedures, and robust peer-review processes (*Pranckutė, 2021*).

Following this, the subsequent step involves formulating search strings. Despite the focus of the SLR on architectural frameworks within data-driven systems in mental health, the initial literature search produced very few relevant studies within the defined scope. As a result, the search was broadened to encompass the health care domain, and the search strings identified were "data-driven", "mental health care architecture", "health care architecture", and "machine learning".

The search strings were used to extract articles published between 2016 and 2023, resulting in 1,011 articles. After removing 14 duplicates, the authors performed an initial screening of titles and abstracts on 997 articles. Of these, 578 articles were excluded as they did not fit the research scope. The remaining 419 articles underwent inclusion and exclusion criteria to identify studies aligning with the objective of this research. This

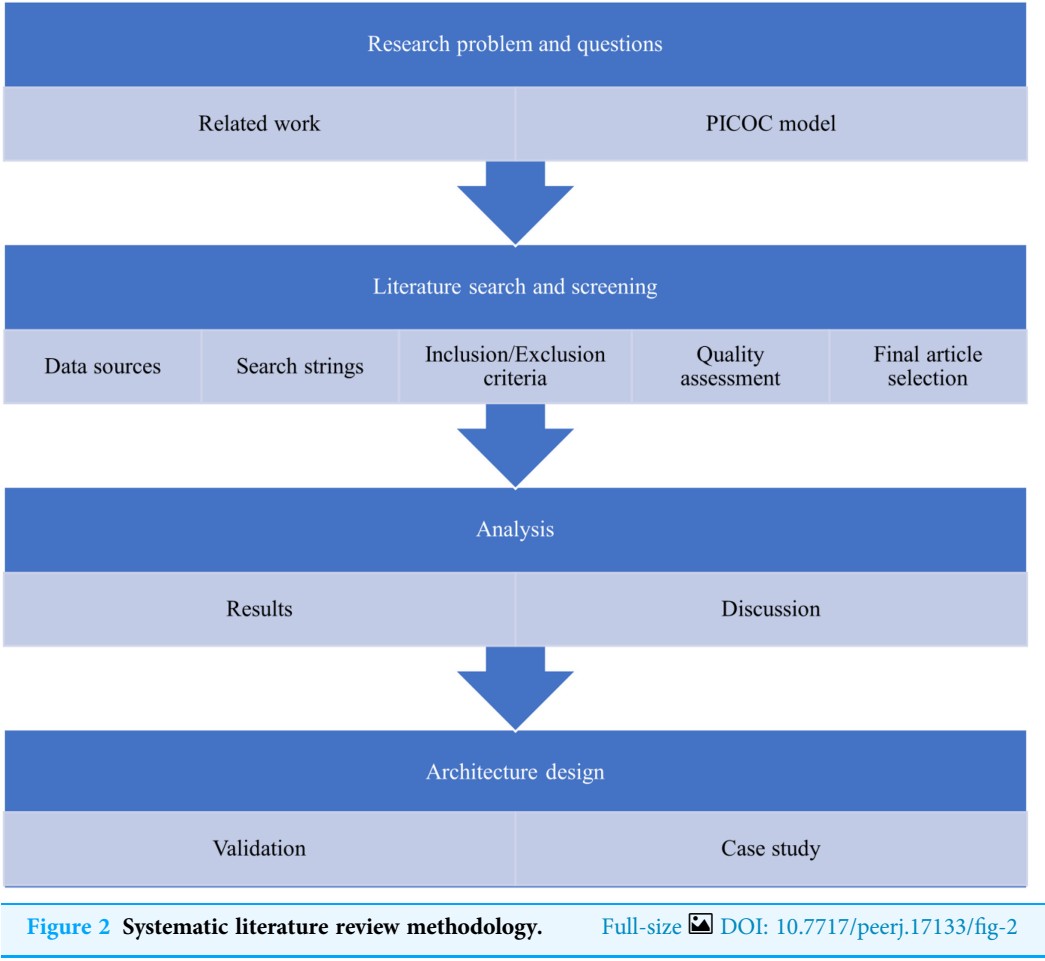

Figure 2 Systematic literature review methodology.

process led to the exclusion of 297 articles, following the criteria outlined in Table 2. The authors then conducted a quality assessment of 122 articles, as detailed in Table 3. Each assessment question was assigned a score of 0–1, where 0 indicates that the article does not meet the quality checklist, 0.5 means it partially meets the checklist, and 1 means it meets the quality criteria. Finally, the authors selected 17 articles with an average score of more than 70% for the final study, as presented in Table 4.

# RESULTS

### RQ1 assessment

To answer RQ1, the review results for each article included in the final study are presented in Table 5, addressing the research questions.

The summary of strengths and limitations features identified in each reviewed study is presented in Table 6 below.

### RQ2 assessment

KEPs are crucial for architecture design as they serve as foundational principles and guidelines, ensuring the effectiveness, efficiency, and reliability of the designed architecture. Figure 3 illustrates the KEPs for the data-driven architectural framework

**Table 2 Inclusion and exclusion criteria.**

| Inclusion criteria | Exclusion criteria |
|---|---|
| The article corresponds to a data-driven architectural framework in the health care domain. | Articles only with machine learning algorithms without any architectural study. |
| The article corresponds to data-driven architecture using machine learning algorithms. | Articles that do not focus on data-driven architecture. |
| Articles including referential architecture for data-driven systems. | Editorial notes, letters, or mini-reviews. |
| Articles in English. | Duplicate publications. |
| Articles that are fully accessible. | Restricted access. |

**Table 3 Quality checklist.**

| Quality assessment | Answer |
|---|---|
| Is there a practical implementation of data-driven architecture? | Yes/Partly/No |
| Does the article cover the machine learning aspect of data-driven systems? | Yes/Partly/No |
| How relevant is the article with respect to research objectives? | Yes/Partly/No |
| Is the research methodology clearly explained? | Yes/Partly/No |
| Are results and analysis referenced with existing works? | Yes/Partly/No |
| Are all the study questions answered? | Yes/Partly/No |

**Table 4 Quality assessment.**

| Index | Q1 | Q2 | Q3 | Q4 | Q5 | Q6 | Total | % |
|---|---|---|---|---|---|---|---|---|
| Shalom, Shahar & Lunenfeld (2016) | 1 | 0 | 0.5 | 1 | 1 | 1 | 4.5 | 0.75 |
| Vinci et al. (2016) | 0.5 | 0 | 1 | 1 | 1 | 1 | 4.5 | 0.75 |
| Schooler et al. (2017) | 1 | 0.5 | 1 | 0.5 | 1 | 0.5 | 4.5 | 0.75 |
| Olsen (2017) | 1 | 0 | 1 | 1 | 1 | 1 | 5 | 0.83 |
| Mendez & Jabba (2018) | 0.5 | 0.5 | 1 | 1 | 0.5 | 1 | 4.5 | 0.75 |
| Chmielewski, Fralszczak & Bugajewski (2018) | 1 | 0 | 0.5 | 1 | 1 | 1 | 4.5 | 0.75 |
| Handayani et al. (2019) | 0.5 | 1 | 1 | 0.5 | 1 | 1 | 5 | 0.83 |
| Beinke, Fitte & Teuteberg (2019) | 1 | 0 | 1 | 1 | 1 | 1 | 5 | 0.83 |
| Zhuang et al. (2020) | 0.5 | 1 | 1 | 1 | 1 | 1 | 5.5 | 0.92 |
| Tummers et al. (2021) | 1 | 0.5 | 0.5 | 1 | 0.5 | 1 | 4.5 | 0.75 |
| Nadhamuni et al. (2021) | 1 | 0.5 | 1 | 0.5 | 1 | 0.5 | 4.5 | 0.75 |
| Ilyas et al. (2022) | 1 | 0 | 1 | 1 | 0.5 | 1 | 4.5 | 0.75 |
| Blobel et al. (2022) | 1 | 0 | 1 | 1 | 1 | 0.5 | 4.5 | 0.75 |
| Aldabbas et al. (2022) | 1 | 0 | 1 | 1 | 0.5 | 1 | 4.5 | 0.75 |
| Perez et al. (2023) | 1 | 0 | 1 | 1 | 1 | 0.5 | 4.5 | 0.75 |
| Mishra et al. (2023) | 1 | 1 | 1 | 1 | 1 | 0.5 | 5.5 | 0.92 |
| Upadhyay et al. (2023) | 1 | 0 | 0.5 | 1 | 1 | 1 | 4.5 | 0.75 |

**Table 5 Review of articles under study.**

| Study | Description |
|---|---|
| *Shalom, Shahar & Lunenfeld (2016)* | • Designed PICARD architecture for continuous clinical guideline-based decision support.<br>• Emphasized interoperability through Application Programming Interface (API) and continuous evaluation with clinicians' input for validation but requires robust data security and privacy measures for critical patient data. |
| *Vinci et al. (2016)* | • Proposed enterprise architecture (EA) based on the specific context of indicators of spatial and temporal dimensions of the mental health care matrix.<br>• Challenges include data availability and accuracy for real-time monitoring. |
| *Schooler et al. (2017)* | • Examined Edge and Fog computing correlation, using data-centric networking to optimize meta-data-driven processes.<br>• Explored software-defined strategies for controlling upstream data flows, including analytics and data caching placement.<br>• Focused on edge and fog computing limits service generalizability to conventional network infrastructures. |
| *Olsen (2017)* | • Focused on implementing EA in the health sector through an exploratory study.<br>• Identified challenges hindering generalized EA, including unclear roles, ineffective communication, low EA maturity, and complex tools.<br>• Attributed challenges to lack of clarity in EA concept, difficult terminology, and complexity of manipulation in EA frameworks. |
| *Mendez & Jabba (2018)* | • Explored IoT health architecture for interoperability using specific communication protocols for heart monitoring.<br>• Identified potential challenges in optimizing system integration and performance. |
| *Chmielewski, Fralszczak & Bugajewski (2018)* | • Achieved key health care application requirements such as data security, authorized access, reliability, efficiency, and context awareness through user authorization, sensor configurations, patient profile management, and wireless transmission.<br>• Noted limitations in exploring health care application interoperability due to data consistency and integration challenges from diverse data sources. |
| *Handayani et al. (2019)* | • Implemented a Health Referral Information System capturing scenarios and viewpoints from stakeholders.<br>• Lacks adequate addressing of significant technical concerns related to data scaling and security in the referral process. |
| *Beinke, Fitte & Teuteberg (2019)* | • Encompassed primary, secondary, and tertiary stakeholders to validate around 30 requirements for a blockchain-based electronic health record to ensure data security and privacy.<br>• Identified major challenges in blockchain health care, including low processing speed, data verification, access and authorization issues, limited scalability, and inadequate computational power. |
| *Zhuang et al. (2020)* | • Primarily addressed the challenge of data coordination, emphasizing security and privacy in health information systems.<br>• Proposed a three-layer architecture utilizing smart contracts in the interfacing layer for facilitating actions between health facilities.<br>• Highlighted limitations, including setup requirements for participating health facilities and potential performance issues related to blockchain nodes generating a massive number of transactions simultaneously. |

(Continued)

| Table 5 (continued) | |
|---|---|
| **Study** | **Description** |
| *Tummers et al. (2021)* | • Designed a Health Information Systems reference architecture using feature modeling with 17 predefined stakeholder viewpoints. |
| | • Employed a comprehensive approach and views for representation and analysis. |
| | • Limited scalability testing across diverse health information system scenarios. |
| *Nadhamuni et al. (2021)* | • Proposed a five-layer enterprise architecture for interoperability and standardization. |
| | • Included a consent layer for individual consent management, aligning with a security strategy. |
| | • Need for further investigation to handle data heterogeneity and to facilitate smooth data exchange among different channels. |
| *Ilyas et al. (2022)* | • Developed a fog-based 4+1 view architecture for real-time data transmission in IoT health care systems to mitigate traffic delay by a decentralized fog computing paradigm. |
| | • Demonstrated enhanced performance with reduced network latency. |
| | • Recognized limitations in security and privacy aspects. |
| *Blobel et al. (2022)* | • Proposed ontology-based generic reference architecture to address multi-disciplinary interoperability challenges in various domains. |
| | • Enables harmonization and mapping of diverse specifications without revisions. |
| | • Offered a policy-driven, system-oriented solution for transforming health and social care ecosystems. |
| | • Lacks a comprehensive investigation into performance analysis within the framework of multidisciplinary integration. |
| *Aldabbas et al. (2022)* | • Designed IoT health care system architecture for real-time patient health data analysis. Focused on personalized drug recommendations using clustering and machine learning. |
| | • Tackled the challenge of data mining data by the use of ensemble models for data analysis and manipulation. |
| | • Identified a gap in addressing security concerns related to cloud-based data transfer. |
| *Perez et al. (2023)* | • Proposed a distributed architecture for an accompaniment service for the elderly and dependents for a streamlined deployment. |
| | • Introduced conceptual, technical, and deployment architectures for implementation and scalability. |
| | • Potential constraints in addressing interoperability due to reliance on various technologies and manufacturers posing challenges in integrating diverse functionalities. |
| *Mishra et al. (2023)* | • Increasing adoption of health care-specific APIs in the health care sector for interoperability and secure data transfer. |
| | • Proposed a framework for enhancing interoperability through the use of API integration in process and system layers. |
| | • Noted shortcomings in addressing data privacy and security concerns during information transfer between health IT systems and third-party applications. |
| *Upadhyay et al. (2023)* | • Explored limitations of an existing system related to sensor data transfer, affected by external noise and error-prone survival tracking techniques using ECG. |
| | • Included the use of machine learning models, enhancing sensor quality, managing data, enabling real-time communication, and improving patient monitoring. |
| | • Emphasized the need for further investigation into the practical implementation and performance of proposed improvements, such as narrowband IoT and scheduling mechanisms. |

**Table 6 RQ1 summary.**

| Study | Strengths | Limitations |
|---|---|---|
| *Shalom, Shahar & Lunenfeld (2016)* | Interoperability | Security and privacy |
| *Vinci et al. (2016)* | Contextualization | Availability & Accuracy |
| *Schooler et al. (2017)* | Performance | Service generalizability |
| *Olsen (2017)* | Consistency | Data manipulation complexity |
| *Mendez & Jabba (2018)* | Interoperability | Performance |
| *Chmielewski, Fralszczak & Bugajewski (2018)* | Security, authorized access, reliability, efficiency, and context | Data consistency and interoperability |
| *Handayani et al. (2019)* | Viewpoint contextualization | Data scaling and security |
| *Beinke, Fitte & Teuteberg (2019)* | Security and privacy | Performance and scalability |
| *Zhuang et al. (2020)* | Security and privacy | Performance |
| *Tummers et al. (2021)* | Viewpoint contextualization | Scalability |
| *Nadhamuni et al. (2021)* | Interoperability and standardization | Data consistency |
| *Ilyas et al. (2022)* | Performance | Security and privacy |
| *Blobel et al. (2022)* | Interoperability, Contextualization, Security, Data manipulation | Performance |
| *Aldabbas et al. (2022)* | Data manipulation | Security and privacy |
| *Perez et al. (2023)* | Scalability | Interoperability |
| *Mishra et al. (2023)* | Interoperability | Data privacy and security |
| *Upadhyay et al. (2023)* | Data exchangeability | Performance |

derived from the review studies to address RQ2. In our proposed approach, the authors have identified six essential elements that constitute the foundation of any data-driven architectural pattern:

1) Data security: Prioritizing data security (*Ping, 2022*) is paramount, encompassing robust authentication, authorization, and the preservation of medical data privacy.

2) Data availability: Ensuring data availability (*Chmielewski, Fralszczak & Bugajewski, 2018*) is pivotal for scalability, performance, and fault prevention. Employing load balancing techniques efficiently distributes data loads.

3) Data quality: Maintaining high data quality (*Rath, Mandal & Sarkar, 2023*) is essential for effective manipulation. This involves addressing missing data through techniques like imputation, ensuring completeness *via* data augmentation, and maintaining consistency through standardizing formats. Data cleaning tackles inconsistencies, outliers, and errors. Validity is further ensured through outlier detection and expert verification, forming the foundation for machine learning manipulations.

4) Data manipulation: Critical for machine learning algorithms, data manipulation facilitates automated prediction and analysis (*Ramirez-del Real et al., 2022*). This includes parameter tuning, feature selection, model selection, and evaluation for machine learning models.

5) Data as a service: Implementing data as a service (*Rahmatulloh et al., 2021*) fosters distributed data management, enhancing interoperability and exchangeability among different applications.
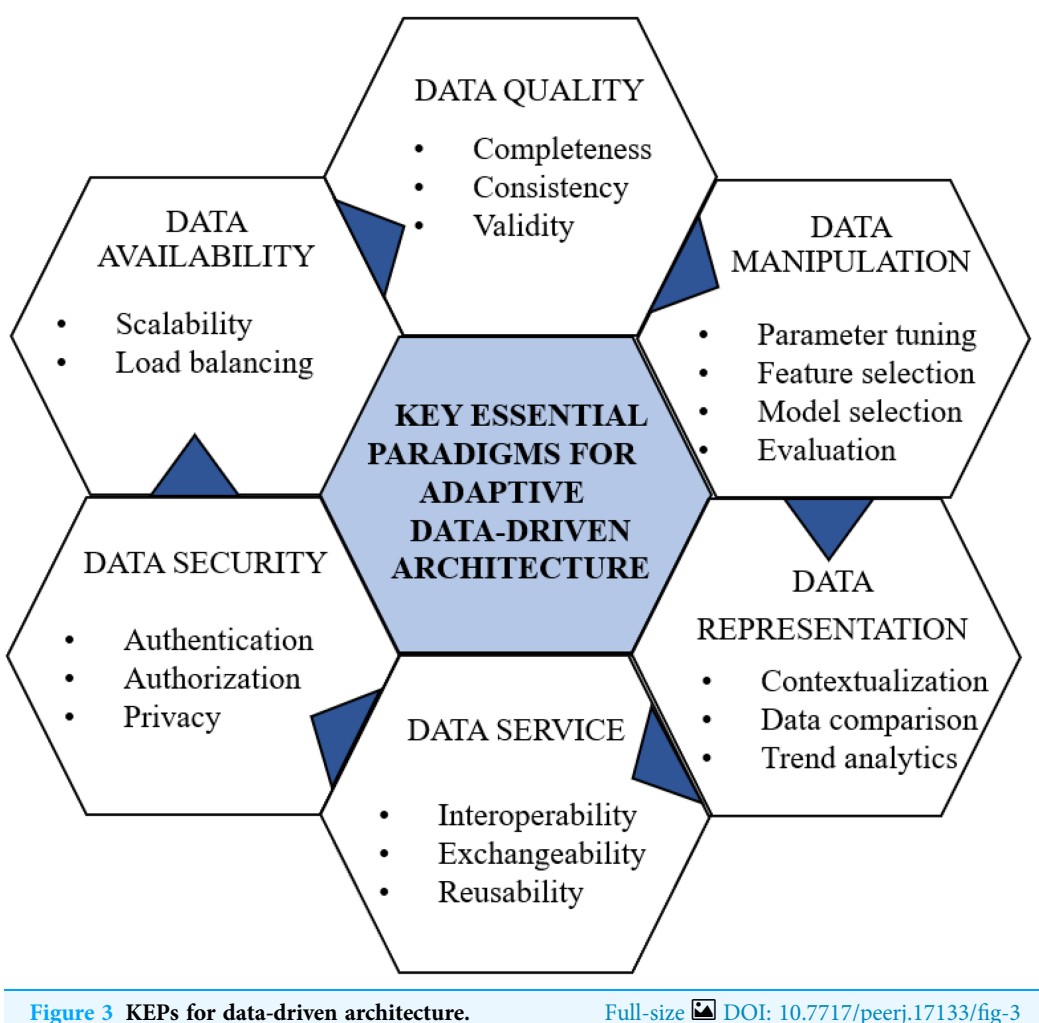

**Figure 3  KEPs for data-driven architecture.**

6) Data representation: Proper data representation is vital for contextualization, enabling insightful comparisons and trend analytics (*Bhavadharini et al., 2019*).

These six KEPs serve as the cornerstone for crafting resilient and efficient data-driven architectures capable of tackling diverse challenges and meeting the requirements of the health care domain.

The mapping matrix in Table 7 illustrates the correlation between RQ1 results and RQ2 KEPs. A checkmark (✓) and crossmark (X) denote the presence of strength or limitation in a particular study for a specific KEP, respectively. This matrix offers a visual representation of how the strengths and limitations identified in each study align with the KEPs.

## DISCUSSION

The strengths and limitations identified in these studies highlight the diverse challenges and factors inherent in health care system architecture, including data interoperability, security, privacy, performance, and consistency (*Ismail et al., 2022*). These findings

**Table 7 Mapping matrix between studies and KEPs.**

| Study | Data security | Data availability | Data quality | Data manipulation | Data service | Data representation |
|---|---|---|---|---|---|---|
| *Shalom, Shahar & Lunenfeld (2016)* | X | | | | ✓ | |
| *Vinci et al. (2016)* | | X | | | | ✓ |
| *Schooler et al. (2017)* | | ✓ | | | X | |
| *Olsen (2017)* | | | ✓ | X | | |
| *Mendez & Jabba (2018)* | | X | | | ✓ | |
| *Chmielewski, Fralszczak & Bugajewski (2018)* | ✓ | | ✓ | | X | ✓ |
| *Handayani et al. (2019)* | X | X | | | | ✓ |
| *Beinke, Fitte & Teuteberg (2019)* | ✓ | X | | | | |
| *Zhuang et al. (2020)* | ✓ | X | | | | |
| *Tummers et al. (2021)* | | X | | | | ✓ |
| *Nadhamuni et al. (2021)* | | | | X | ✓ | |
| *Ilyas et al. (2022)* | X | ✓ | | | | |
| *Blobel et al. (2022)* | ✓ | X | | ✓ | ✓ | |
| *Aldabbas et al. (2022)* | X | | | ✓ | | |
| *Perez et al. (2023)* | | ✓ | | | X | |
| *Mishra et al. (2023)* | X | | | | ✓ | |
| *Upadhyay et al. (2023)* | | X | | | ✓ | |

provide valuable insights for software architects and practitioners involved in developing resilient and efficient data-driven health care systems.

Furthermore, the mapping matrix reveals deficiencies in critical elements of data-driven architecture across each study, emphasizing the need for an SLR to discern KEPs for designing an adaptive, data-driven architecture characterized by comprehensive robustness.

Designing an adaptive data-driven architecture based on KEPs involves creating a flexible and responsive system that can dynamically adjust and optimize its processes based on mental health care requirements. The KEPs identified in Fig. 3 serves as a guiding principle for shaping the architecture's structure and functionalities. The adaptive data-driven architecture for mental health care application is presented in Fig. 4, with KEPs distributed across the designed architecture as follows:

Data security holds paramount importance in every organization, particularly in mental health care, where handling sensitive and personal information is crucial (*Fysarakis et al., 2019*). Two key measures, role-based access control (RBAC) and multi-factor
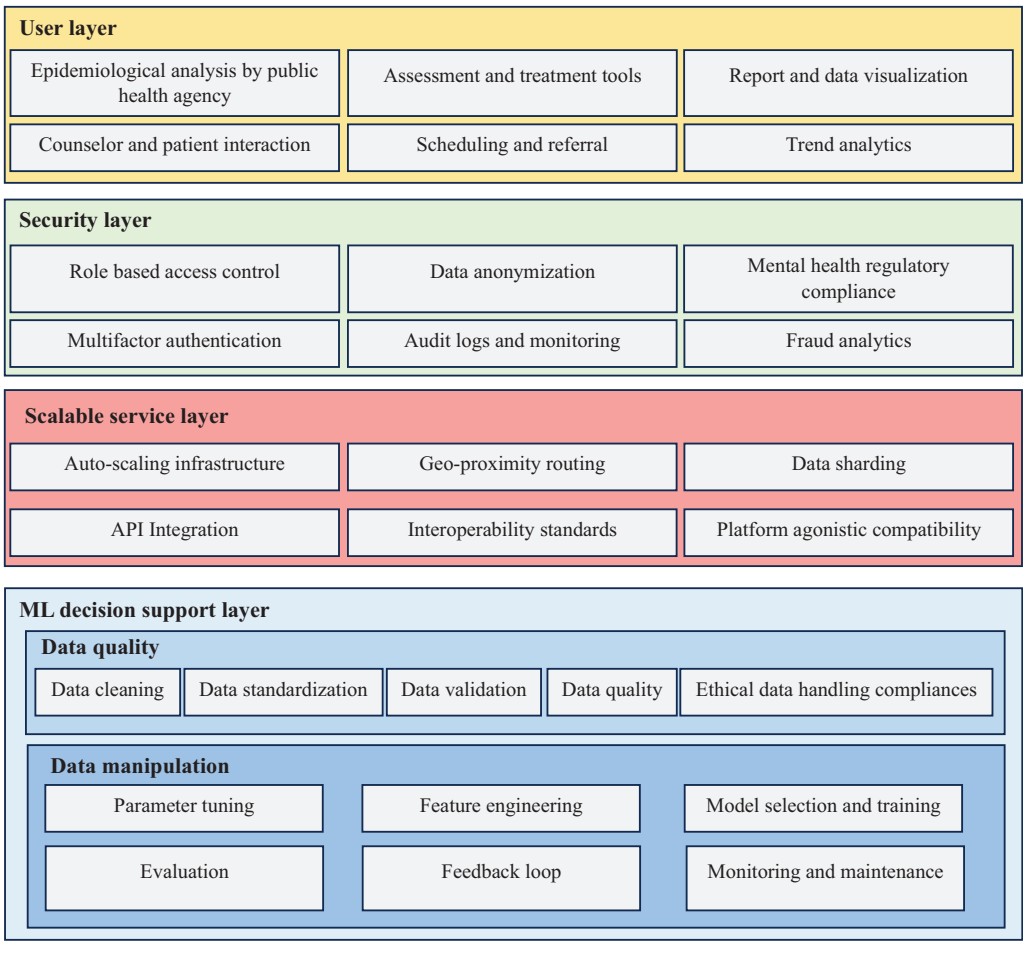

**Figure 4  Adaptive data-driven mental health care architecture.**

authentication (MFA), play pivotal roles in safeguarding sensitive data. RBAC enables mental health care organizations to assign specific access privileges to users based on their roles and responsibilities (*Garg et al., 2023*), preventing unauthorized access and ensuring users only access necessary information for their job functions. MFA enhances security by requiring users to provide multiple forms of authentication, such as using hash functions (*Midha et al., 2023*). Data anonymization (*Tomás, Rasteiro & Bernardino, 2022*) is another critical aspect of data security. Removing or encrypting personally identifiable information from datasets protects individuals' privacy while enabling data analysis and research. However, maintaining data utility during anonymization is a challenge, as overly aggressive anonymization can compromise data integrity and research potential. Compliance with relevant laws and regulations is essential for mental health organizations to protect patient confidentiality and avoid legal repercussions (*Gooding & Kariotis, 2021*). Audit logs and monitoring play a crucial role in detecting and responding to potential security breaches (*Yang et al., 2021*). By monitoring access logs and system activities, organizations can identify suspicious behavior and take prompt action (*Ali, Ahmed &*

*Khan, 2021*). Fraud analytics, an emerging field, utilizes advanced data analytics techniques to detect and prevent fraudulent activities (*Ai et al., 2022*). Applying machine learning algorithms and anomaly detection methods to transaction data enables mental health organizations to proactively identify and mitigate fraudulent behavior.

Ensuring data availability is paramount in mental health care to guarantee timely access to critical information. The implementation of auto-scaling infrastructure enables the system to dynamically allocate resources based on demand (*Santos, Paulino & Vardasca, 2020*). This ensures the system's capability to handle varying workloads and maintain high availability, even during peak times. Geo-proximity routing, a technique routing data traffic to the nearest data center or server based on the user's geographical location, is employed to reduce latency and ensure faster data access for users in different regions (*Meena, Gorripatti & Praba, 2021*). Additionally, data sharding, involving the horizontal partitioning of data across multiple servers or databases, enhances data retrieval efficiency and scalability (*Niya, Beckmann & Stiller, 2020*).

Maintaining data quality is crucial for accurate analysis and informed decision-making in mental health care. Data cleaning involves identifying and correcting errors, inconsistencies, and missing values in the dataset (*Sorkhabi, Gharehchopogh & Shahamfar, 2020*). Data standardization is another vital step, ensuring that data is transformed into a consistent format and structure. Standardized data (*McMullen et al., 2022*) allows for easier integration and analysis across different systems. Concurrently, data validation ensures that data meets specific criteria and aligns with predefined rules. Implementing automated validation checks aids in identifying data discrepancies and upholding data accuracy, though it necessitates regular updates to adapt to evolving data requirements. Compliance with ethical data handling principles (*Wilton, 2017*) involves adhering to ethical standards and regulations related to data collection, usage, and storage. Mental health care organizations must prioritize aspects such as data privacy, informed consent, and data anonymization to safeguard patient rights.

Data manipulation forms a critical aspect of developing machine learning models. This involves several key steps. Parameter tuning (*Jebeile et al., 2023*) which fine-tunes the model parameters to optimize performance. Feature engineering focuses on selecting or creating relevant features to enhance model accuracy and predictive power. Model selection and training involve choosing the most suitable machine learning algorithm and training the model on the dataset. Evaluation is the process of assessing the model's performance on a separate test dataset to measure its accuracy, precision, recall, and other metrics (*Nithya & Ilango, 2017*). The feedback loop enables continuous improvement by incorporating new data and refining the model based on mental health professional feedback and real-world scenarios. Additionally, monitoring and maintenance are crucial to ensure the model remains accurate and up-to-date over time. Regular monitoring helps identify concept drift and data changes that might affect model performance, while maintenance involves periodic retraining and updates.

Data service is centered around making data accessible and usable across diverse platforms and systems. Key considerations include platform-agnostic compatibility (*Barron-Lugo et al., 2023*), ensuring that data services are not tied to any specific platform,

**Table 8 Expert review analysis.**

| Element | Validating criterion (Feature) | Mean value | Confidence intervals |
|---|---|---|---|
| Maturity levels | Maturity levels are compliant at all stages of data processing (Sufficiency). | 4.4 | [4.1, 4.7] |
| | Is data redundancy effectively reduced and noise minimized? (Accuracy) | 4.0 | [3.6, 4.4] |
| Process and practices | All processes and practices are generalizable to domain-relevant applications (Relevancy). | 4.3 | [3.9, 4.7] |
| | It encompasses all processes impacting the domain (Comprehensiveness). | 4.6 | [4.3, 4.9] |
| | Process and practices are analyzed and distinct (Mutual Exclusion). | 4.4 | [4.1, 4.7] |
| Ease of use | Is the architectural framework easy to comprehend? (Understandability) | 4.1 | [3.6, 4.6] |
| | Is the architecture feasible for implementation? (Feasibility) | 4.7 | [4.4, 5.0] |
| Applicability | Is it practically applicable to the mental health care domain (Applicability) | 4.4 | [4.1, 4.7] |
| Flexibility | Is it adaptable to different mental health care systems (Adaptability) | 4.7 | [4.3, 5.1] |
| | Technical flexibility to integrate different technologies (Technical flexibility). | 4.5 | [4.2, 4.8] |
| | Is it flexible enough to be customized for accommodating changes? (Change flexibility) | 4.2 | [3.7, 4.7] |

and allowing seamless integration with various devices and applications. Interoperability standards facilitate data exchange and communication between different systems (*Fysarakis et al., 2019*). Adopting common data formats and protocols ensures smooth data flow and reduces integration challenges. API (*Uddin, Khomh & Roy, 2020*) Integration enables data services to interact with external applications and services, promoting collaboration and information sharing across different platforms.

Data representation plays a pivotal role in conveying insights and information to stakeholders. Report and data visualization (*Fallon & Crouser, 2019*) present data in a visually appealing and easy-to-understand manner, making it simpler for users to interpret complex data. Epidemiological analysis involves analyzing and interpreting health-related data to identify patterns and trends (*Kakalou, Lazarus & Koutkias, 2019*), facilitating evidence-based decision-making in mental health care. Trend analytics enables the identification of long-term patterns and developments in data, allowing organizations to make informed predictions and strategic decisions. Counselor and patient profiles store important information about mental health professionals and patients, supporting personalized and effective treatment plans. Scheduling and referral features (*Guddu & Demissie, 2022*) help manage appointments and referrals efficiently, ensuring smooth coordination among health care providers and optimizing patient care. Assessment and treatment features assist mental health professionals in evaluating patients' conditions and delivering appropriate treatments based on data-driven insights.

After completing the architectural design, the authors engaged with ten professional experts to evaluate the adaptive data-driven architecture for mental health care. The assessment instrument focused on maturity levels, processes and practices, ease of use, applicability, and flexibility. Table 8 compiles the validation criteria, covering 11 features that were used during the expert review session, along with the mean value of each criterion. The evaluation utilized a 5-point Likert scale (*Paes et al., 2021*; *Perez-Benito et al., 2020*; *Tungpantong, Nilsook & Wannapiroon, 2022*; *van den Bergh et al., 2020*) to record the feedback from experts.

The evaluated features represent crucial factors considered within the context of the system or application under scrutiny. Each feature is assigned a numerical mean value on a scale from one to five, where one denotes a 'strongly disagree' level, and five indicates a 'strongly agree' level. The confidence intervals provide a range within which the true population mean is likely to fall. In this analysis, a higher value indicates a stronger agreement with the given criterion. Upon scrutinizing the results, it is evident that the mean values collectively suggest a well-rounded performance of the assessed elements across various validating criteria. Noteworthy elements are evident in criteria such as feasibility, adaptability, comprehensiveness, technical flexibility, sufficiency, applicability, and mutual exclusion where the mean values are notably high in range of 4.4 to 4.7. This implies that, commendable performance by the elements in these specific aspects. Additionally, criteria with such as relevancy, change flexibility, understandability, and accuracy with mean values in range 4.1 to 4.3, indicate areas for further improvement. In light of unified feedback from experts, it is recommended to proceed with the real-time implementation of the proposed architecture to assess and refine its capabilities for deployment adaptation. As an initial phase for this real-time deployment, the article outlines the design of a conceptual prototype to implement our devised architecture for monitoring anxiety disorders during a pandemic event.

## CASE STUDY

The Pandemic Anxiety Prediction Application (PAPA) was prototyped based on the proposed adaptive data-driven mental health care architecture. During a pandemic, anxiety disorders become a major concern, necessitating the application of machine learning predictions to a large population sample (*Albagmi et al., 2022*). In the event of a pandemic, the government could employ PAPA to trigger mental health screening for vulnerable populations in targeted high-risk areas. PAPA utilizes machine learning anxiety prediction models to estimate anxiety levels and recommends online or nearby mental health counseling services. It also keeps track of mental health analytics, aiding the government in data-driven decision-making. The basic process flow diagram of PAPA is provided in Fig. 5. Data sources for PAPA include online datasets, anxiety assessment instruments such as the Generalized Anxiety Disorder-7 (*Camargo et al., 2021*), and reports from mental health care centers. Categorizing these sources facilitates rendering the data as an independent service for other functionalities, promoting data exchangeability and interoperability between government health care institutions for a referral.

Data security is ensured through multi-factor authentication, with authorization achieved through restricted user role-based access for government entities, respective health care professionals, and users. Privacy is maintained through encryption mechanisms (*Ping, 2022*) and consent management for data storage. User identity is also anonymized for the machine learning training dataset. Data is periodically archived and backed up on cloud servers for scalability (*Quinde et al., 2020*) and load balancing. Standard processes of machine learning algorithms, such as data cleaning, preprocessing, parameter tuning, feature (*Mounica & Lavanya, 2022*) and model selection and prediction,

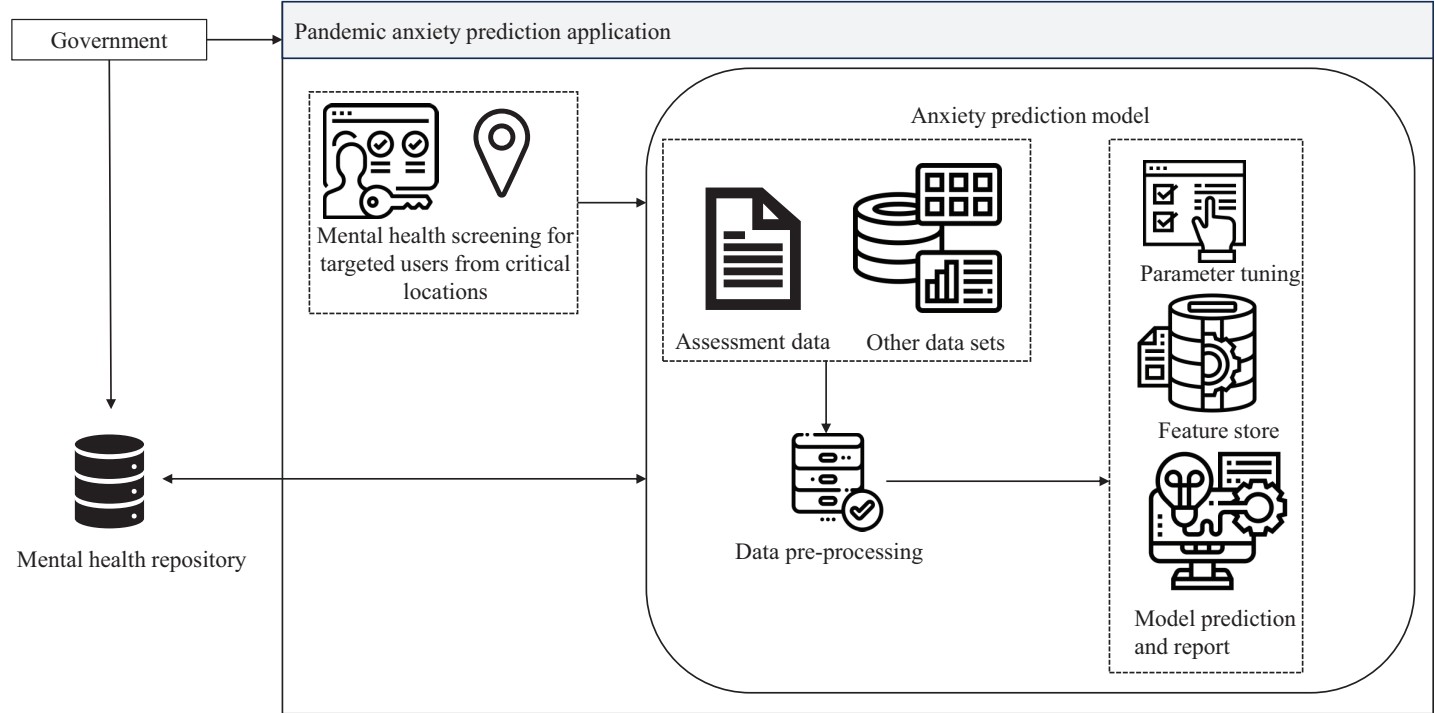

**Figure 5** Process flow of pandemic anxiety prediction application.

are applied in the anxiety prediction model. The anxiety prediction model is continuously trained based on participant assessment responses and social media data, such as tweets and Facebook posts. This application has the potential to aid in the early identification of individuals experiencing pandemic-related anxiety. Early prediction allows for timely intervention and support to mitigate the impact on mental health (*Riepenhausen et al., 2022*). Additionally, accurate prediction of pandemic anxiety levels could help allocate mental health resources more efficiently, directing support to those who need it the most during periods of heightened anxiety. By predicting pandemic anxiety levels for individuals, mental health care providers could offer more personalized and tailored treatment plans to address specific needs and coping mechanisms.

## CONCLUSION

In conclusion, the adaptive data-driven architecture for mental health care applications represents a transformative leap towards personalized, efficient, and compassionate health care services. By harnessing the power of data analytics, machine learning, and advanced technologies, this architecture empowers mental health professionals and government agencies with data-driven insights and evidence-based decision-making capabilities. The main goal of this study was to investigate the prominent strengths and limitations encountered in current health care software architectures. The research revolves around formulating an adaptive, data-driven architecture capable of effectively mitigating these identified limitations and enhancing strengths using key essential paradigms. The adoption of an adaptive data-driven architecture for mental health care applications could

have several significant impacts, like personalized treatment to provide personalized treatment plans considering patients' profiles and responses to interventions, leading to more effective and tailored mental health care. The architecture's adaptability enables real-time monitoring of patients' mental health indicators through assessment tools. In cases of deterioration, timely interventions can be initiated, preventing potential harm and ensuring prompt support. By analyzing large-scale mental health data, the adaptive architecture can identify patterns and trends that may contribute to improved diagnostic accuracy and prognosis for mental disorders. Its ability to optimize resources and treatment plans can lead to cost-effective mental health care, reducing unnecessary expenditures and maximizing the impact of interventions. With personalized customization from the adaptive architecture and remote mental health care options, it can help reduce stigma and access barriers. Patients can access support discreetly, enhancing mental health care utilization. Overall, an adaptive data-driven architecture in mental health care applications has the potential to revolutionize the delivery of mental health services, providing more effective, personalized, and accessible care to individuals in need. The proposed architecture underwent a thorough evaluation by subject matter experts, which confirmed its adaptability to diverse mental disorders. The radar values provided valuable insights into the strengths and limitations of the proposed architecture concerning the identified features.

The high mean values for most constraints demonstrate the effectiveness of the system design in addressing these factors. However, the disparities in mean values prompt further analysis and refinement to ensure the architecture's robustness. By leveraging these insights, application designers and stakeholders can make informed decisions to fine-tune the architecture, ultimately enhancing its adaptability and effectiveness in dealing with various mental disorders. However, it is important to note that the review's scope was limited to peer-reviewed journal articles accessible through the databases indexed in Web of Science. Therefore, it did not encompass other types of publications, which could be considered a limitation of this study. Moving forward, the future scope of research involves in development and real-time deployment of pandemic anxiety prediction application, aiming to identify the technical challenges associated with the implementation of the proposed adaptive architecture. This would provide valuable insights into refining and enhancing the software architecture to effectively address real-world scenarios and improve mental health care.

## ACKNOWLEDGEMENTS

Special thanks to the professional experts whose valuable contributions were instrumental in validating the proposed architecture.

### Funding

This work was supported by the Ministry of Higher Education Malaysia for the Fundamental Research Grant Scheme (FRGS/1/2022/SS09/UM/02/4). The funders had no

role in study design, data collection and analysis, decision to publish, or preparation of the manuscript.

## Grant Disclosures

The following grant information was disclosed by the authors:
Ministry of Higher Education Malaysia: FRGS/1/2022/SS09/UM/02/4.

## Competing Interests

The authors declare that they have no competing interests.

## Author Contributions

- Aishwarya Sundaram conceived and designed the experiments, analyzed the data, prepared figures and/or tables, authored or reviewed drafts of the article, and approved the final draft.
- Hema Subramaniam conceived and designed the experiments, analyzed the data, prepared figures and/or tables, authored or reviewed drafts of the article, and approved the final draft.
- Siti Hafizah Ab Hamid performed the experiments, authored or reviewed drafts of the article, and approved the final draft.
- Azmawaty Mohamad Nor performed the experiments, authored or reviewed drafts of the article, and approved the final draft.

## Data Availability

This is a systematic literature review.

## Supplemental Information

Supplemental information for this article can be found online at http://dx.doi.org/10.7717/peerj.17133#supplemental-information.

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
