# Peer review of "An adaptive data-driven architecture for mental health care applications"

_PeerJ, doi:10.7717/peerj.17133_

## Round 0.1 · original submission · Major Revisions

Dear Authors,

Thank you for submitting your manuscript on this important topic. I'd like to endorse the reviewers' comments on how this manuscript would improve substantially: a major criticism is the vaguely defined objective and the limited usage of database for your literature research. Such issues are essential for a useful contribution to the literature. Furthermore, the article as a whole should be more concise, provide more references, and follow a more stringent red line. However, due to the importance of this field and the implications that can be drawn, I would endorse the publication, if you can provide a revised manuscript with the suggested changes and depending on the reviewers' comments.
Kind regards,
Johanna Löchner

**Language Note:** The review process has identified that the English language must be improved. PeerJ can provide language editing services - please contact us at copyediting@peerj.com for pricing (be sure to provide your manuscript number and title). Alternatively, you should make your own arrangements to improve the language quality and provide details in your response letter. – PeerJ Staff

Reviewer 1 ·

Basic reporting

- English language correctness and readability
* Please review the paper (especially the Abstract and the Case Study) regarding language correctness (e.g., using Grammarly or a fluent English-speaking person).
* Please check your figure and tables for language correctness (e.g., Figure 2: 'summarize', Table 1: 'strengths').
* The readability of the 'Introduction' and the 'Background Study' would benefit from being divided into paragraphs at appropriate places.
* Do not introduce acronyms that are never used (e.g., DDSI) or only used once (e.g., DDDM, CPS), this is confusing and makes it harder to read.
* Grammar and spelling:
** l49: The point before the references should be removed.
** l56: The correct naming (according to the WHO) of the pandemic is 'COVID-19'.
** l92: Something went wrong with the grammar here, maybe "This study aims to design an adaptive data-driven architecture, ...".
** l513: ins -> are

- Structure and easiness to follow the report
* The paper lacks a clear red thread and is thus not easy to follow. A couple of points that could help are:
** The structure of the report is not clearly explained anywhere. A paragraph indicating what each section is about at the end of the introduction could help the reader to know what to expect.
** l85f does not follow clearly from the previous part. It feels like this is the result of the literature review which follows afterwards.
** Please rename the chapter 'Background Study' to 'Literature Review' or 'Related Work', as the current title is misleading, especially combined with the first sentence: 'This study aims on design of an adaptive data-driven architectures, which focuses to conduct a systematic review to identify key architectural strengths and limitations in this domain.' Readers might be confused if this is already part of the review study.
** In the section 'Background Study' it is hard to follow which information belongs to which citation (e.g., l100-l111). Please cite the source after the very first sentence and make it clear you are still talking about the same work in your subsequent sentences (e.g., using 'they').
** l52-l178: Right now it is not clear that this is only an introduction for the following chapters, leaving the reader to think that information is missing and confused about why the research problem and questions come after the methods.
* The chapters 'Literature search and screening' and 'Extraction and data analysis' are not clearly separated.
** The inclusion and exclusion criteria as well as the screening results should go into the 'Literature search and screening' chapter.
** Quality assessment should be in the section about screening, as it is in Figure 1.
** The part about how the actual data is extracted is currently in the chapter about the research problem.
* What is the chapter 'Case Study' about? It was not mentioned anywhere before. How is this connected to everything else? Why was this done? How was it done? Where is the discussion about this? This chapter seems to float there without a connection to the other parts.

- Figures and tables
* Figure 1 uses an acronym that has not been introduced in the text prior (WoS).
* Which additional information is Figure 3 providing? If the sentence in l190f is changed to 'The process of mapping strengths and limitations to derive KEP encompased four steps:' it is entirely redundant.
* Table 4 uses indexes for the studies, which leaves it unclear which evaluation corresponds to which study.
* Figure 6 is of insufficient quality making it extremely difficult to read.
* What information does Figure 7 provide which is not included in Table 7? What is the added value of including this figure?

- Other
* Why is l243 highlighted?

Experimental design

- Research question
* The knowledge gap does not follow clearly from the literature review right now. For example: Which recent trends highlight the need for an adaptive architecture, as written in l148?
- Literature search
* l209ff: Please put quotation marks around the search strings. Right now it is not clear if 'data-driven' was a search string by itself for example.

Validity of the findings

- RQ 1
* The display of results is in the form of a three-page text, which is hard to follow and makes the quick deduction of information and comparison between studies impossible. Maybe it would be possible to organize this information in an appropriate table or as bullet points.
* The system of how the text is divided into paragraphs does not reveal itself to me.
- RQ 2
* Please do not indicate the existence of strengths and weaknesses in Table 6 using the same symbol. This is completely misleading and does not allow 'for a clear visualization of the strengths and limitations identified in each study and how they are mapped to KEP' as you put it.
* Please remove the 'Total key essential paradigms per article' in Figure 5. There is no additional information gained from this and it distorts the results as well as makes the graph harder to read.
* Giving three decimal points on a 5-point-Likert scale which only involves the review of ten people suggests an accuracy which is not given (Table 7).
* Figure 7 distorts the real differences by showing only the scale between 3.8 and 4.8. This makes the differences seem bigger than they are, especially considering that they are only compiled from the answers of ten people.
* To analyze the difference in the features results it would be interesting to provide confidence intervals in Table 7 and Figure 7.
* l524ff: I do not understand how the authors come to these conclusions, especially as all values fall between 4.0 and 4.7.
Results and discussion
- Discussion and results
* It is not clear where the result part ends and the discussion begins. They seem to be a little intertwined making it difficult to distinguish what is a result and what might be critical points that need to be further addressed.
* A critical discussion of the expert interview is completely missing, as is a methodology part for this (Who were the experts? How did these review sessions work?).

Additional comments

* The review deals with an important topic, however, in its current state, it lacks a clear red thread, in parts making it impossible to follow what the authors want to convey. In many places, the chapter titles and the information provided do not fit together. I would suggest reorganizing the provided information into the appropriate chapters.

·

Basic reporting

Citations:
A lot of statements in the manuscript are not adequately documented with sources. Especially in the introduction and the background, the citations to a lot of statements are missing. Please provide citations for every statement! In addition, the citations in the text should be done in a proper citation style. Due to the mental health context, a citation according to APA style is recommended. Especially concerning the use of multiple sources and citations in a sentence, corrections should be made, e.g. “(Patel & Gandi 2018) focused on...,” should not be cited in brackets due to the rules and regulations. In addition, the reading flow is disturbed in this way. Moreover, in the method section, the choice of the applied method could be better derived from existing literature.

Language:
Overall, the paper is well written, but there are some grammatical errors, e.g., in the choice of plural/ singular or tenses (e.g. of an adaptive data-driven architectures; an initial screening is done... articles were included). This should be thoroughly checked and revised.
In addition, attention should be paid to an adequate choice of words, e.g. the terms mental health issues or mental health disorder are used. I would recommend using the word mental disorder to also be in line with the DSM-5. In addition, the wording should be used more consistently here. This would be important as there are differences between actual mental disorders and other aspects in this context, like bad mood for example.
Please use the same wording conclusively throughout the manuscript, e.g., Big Data with capital letters. However, the term “Big Data” is controversial and rather refers to datasets such as stock market trading with TerraBits in a few units. “Complex data” is currently a better term.

Introduction and Background:
It might be interesting for the reader to elucidate in more detail the literature on machine learning in the context of mental health. For example, machine learning models in the context of mental health could be described in more detail instead of referring only to the COVID-19 pandemic. In addition, the background study part could be better structured. With a clearer separation of results from other studies and ideas of this manuscript, this part could become more comprehensible. Please sharpen the background towards the mental health application use case.
In this way the research question will be clearer. In my opinion, it is not clear whether it refers to machine learning models and data-driven architectures in general or only those in the context of mental health. From my point of view, it is not completely clear whether the research question is limited to the mental health field or not. The title and the beginning of the introduction point to this goal, and afterwards it becomes more general again. Therefore, this should be specified here at the latest.

You are stating, that “The initial stage of the data collection process for decision-making involves gathering data from sources like social networking sites (e.g., Facebook, Twitter) (Alharbi & Fkih 2022), existing datasets acquired through surveys, cohort studies, etc., or through direct face-to-face interviews, among other methods.”. However, here you are lacking, that data collection relies often on passive sensing, which means gathering data not only from social media but also from ubiquitous mobile devices in everyday life. This data can also be used to train machine learning algorithms. This is important to describe here as well.

You say “It should promote reusability, automated machine learning prediction and human decision support (Alreshidi & Ahmad 2019).” However, please see Terhorst et al. 2023 on Decision Support Systems, which states that the main issue is that we currently cannot have stand-alone decision support systems and that the decision must continue to be made by the human being. However, in addition to reusability, explainability is particularly important.

Experimental design

Methods:
In the PRISMA flowchart it is essential to also report the exclusion reasons for those articles, that were excluded after full-text screening. Please add this.
An extension of the literature search to other databases (e.g. medline, embase, ebsco, pubmed, and specific technological databases such as IEEE) is strongly recommended. To achieve a minimum level of certainty that relevant publications could be identified and to reduce publication bias, at least five databases should be considered.

Validity of the findings

Results and Discussion:
There is no concise summary of results in the beginning of the discussion. Instead, results and discussion are reported together. I would have liked to see a clearer summary of the results here, instead of presenting the results from individual studies. This would make the results section clearer, more understandable, and could put more focus on the answer to the research question. This could be supported, by the usual separation of the result section and its discussion. In addition, it is not clear what the objective of the expert review is. To determine the validity of the data-driven architecture, a validation study would have to be carried out.
Case Study: If the case study shall be part of this manuscript, this must be embedded in the method and results section. It’s confusing for the reader, cause it is not clear, which part it has in the systematic review.

Additional comments

Please don’t use abbreviations in the abstract (e.g., you mentioned WoS as used database; however, please state the whole name of the database instead).
Moreover, please introduce abbreviations before the first use. E.g., Preferred reporting items for systematic reviews and meta-analyses = PRISMA. However, PRISMA guidelines cannot “inspire” but rather you should follow them.
Numbers under twelve should be written in text as a word (e.g., in the abstract you say “…above 4 for each assessed feature,…”).
Figure6 has a quite low quality. Please provide it in a higher resolution.

---

## Round 0.2 · Minor Revisions

Dear Dr. Sundaram,

Your manuscript still requires some Minor Revisions.

Kind regards,
Johanna Löchner

Reviewer 1 ·

Basic reporting

- Spelling, Grammar, Punctuation
* l51: missing period
* l146, l163: space after "et al." missing
* l206f: "interoperability" and "Data service" have other quotation marks than anywhere else in the paper

- Figures and Tables
* There are six questions in Table 3, but the answer options are only listed five times.
* Table 7 seems to break words in the middle without proper hyphenation.
* The checkmarks in Table 7 look like the root sign and are not centred.
* What is the additional value of Figure 4 compared to Table 7? Maybe if the cells in the Table were coloured, the Figure could be discarded?

- Other
* l309: This citation seems to be listing the author's first name even though another person with the same last name does not seem to appear.
* The citations in l442f should be in one bracket.
* The paper is generally written in the third person. Only in very few places(l236, l249, l251) do the authors suddenly use the words "we" and "us". It might be an idea to change this to make it consistent.

Experimental design

* How many experts participate in the rating? I remember from the first version that it was ten; however, that information is nowhere to be found anymore.

Validity of the findings

* l456: The result should only be displayed with one decimal place.
* The confidence intervals in Table 8 should not use more decimal places than the mean value; this suggests an accuracy that is not present.
* l456ff: Why is "applicability" not in this cluster? It has the same mean value as "mutual exclusion". Also, "feasibility" and "technical flexibility" are in the same range. It is unclear what the authors want to express with the cluster anyhow. Why does it matter that they have similar mean values? Especially as all values are relatively close to each other. The same is true for the moderate mean values later in the same paragraph.

Additional comments

Dear authors,
I like how you incorporated many of the suggested changes. The paper reads much more stringent now.

One thing that kind of irritates me, however, is the title. "A systematic review" does not seem to fit the research aim "to create a flexible mental health care architecture" as mentioned in the abstract.

---

## Round 0.3 · accepted · Accept

Thank you for including the recommended changes and the revised manuscript. I'm happy to inform you that we will accept your manuscript for publication. Thank you for sharing your research with us.

Kind regards,
Johanna Löchner